# Intimate Partner Violence in Relation to Husband Characteristics and Women Empowerment: Evidence from Nepal

**DOI:** 10.3390/ijerph16050709

**Published:** 2019-02-27

**Authors:** Sujan Gautam, Hyoung-Sun Jeong

**Affiliations:** Department of Health Administration, Graduate School, Yonsei University, Wonju, Gangwon-do 26493, Korea; gautamsujan@gmail.com

**Keywords:** intimate partner violence, marital control behavior, wife beating, husband characteristics, women empowerment, Nepal, NDHS 2016

## Abstract

The purpose of this study is to assess the magnitude of intimate partner violence (IPV) and associated factors among women in Nepal. The secondary data from the Nepal Demographic and Health Survey (NDHS) 2016 was used. This study was confined to the respondents selected for the domestic violence module. The association between experience of IPV ‘ever’ and ‘in the past year’ with selected factors were examined by using Chi-square test, followed by multivariate logistic regression. Complex sample analysis procedure was adopted to adjust for multi-stage sampling design, cluster weight, and sample weight. The result revealed that 26.3% of ever-married women experienced any form of IPV at some point in their lives, while only 13.7% has experienced any form of IPV in the past year. The factors associated with both ‘lifetime’ and ‘past year’ experience of IPV includes women witnessing parental violence during their childhood, the husband being drunk frequently, women being afraid of their husband most of the times, and women whose husbands shows marital control behavior. Women’s experiencing IPV was associated more with husband related factors than with women’s empowerment indicators. Reducing IPV requires a commitment to changing the norms that promote the husband’s behavior of controlling his wives and beating her.

## 1. Introduction

The World Conference on Human Rights, 1993, recognized violence against women as a violation of human rights and contributed to the Declaration on the Elimination of Violence against Women, 1993. According to the UN, violence against women is defined as; “any act of gender-based violence that results in, or is likely to result in, physical, sexual or psychological harm or suffering to women, including threats of such acts, coercion or arbitrary deprivation of liberty, when occurring in public or in private life” [1]. Intimate partner violence (IPV) refers to “any behavior by an intimate partner that leads to physical, sexual or psychological harm, including physical aggression, sexual coercion, psychological abuse, and controlling behaviors” [2]. IPV as a subset of gender-based violence against women is often treated as domestic violence under the law. In 2009, Nepal passed the 2008 Domestic Violence (Offence and Punishment) Act, and in 2010, the Domestic Violence (Offence and Punishment) regulation. Further, the new constitution of Nepal 2015 (part 3), has guaranteed equal rights for women, and protection against any forms of violence [3]. Goal 5 of Global Sustainable Development Goals, “Achieve Gender Equality and Empower All Women and Girls” calls for the end of discrimination, and eliminate all forms of violence against all women and girls [4].

### 1.1. Global Prevalence of IPV

IPV against women is now well recognized as public health and human right problem and is a manifestation of gender inequality that affects physical and psychological well-being. Most of the statistics known about violence against women come from population-based surveys and special studies. The estimates indicate that the proportion of ever-partnered women reporting to have experienced physical or sexual IPV at some point in their lives was about one in three women worldwide [5], about 37% in African, Eastern Mediterranean and South-East Asia region, about 30% in the region of the Americas, and about 25% in the European and Western Pacific regions [6]. According to a Word Health Organization (WHO) multi-country study on women’s health and domestic violence done in 10 countries, the reported lifetime prevalence of physical and/or sexual IPV ranges from 15% (Japanese city) to 71% (Ethiopian province), and the reported past year prevalence ranged from about 4% (Japanese city, and Serbian and Montenegrin city) to 54% (Ethiopian province) [7].

### 1.2. Prevalence of IPV in Nepal

Various cross-sectional studies done in different parts of Nepal reported women’s lifetime experience of sexual violence by their husband ranged from 16.6–58%, whereas, 7.1–31% of women reported to have experienced sexual violence in the past year [8,9,10]. Women can be a victim of violence at any point in their life. There is also evidence that women have experienced some forms of violence during pregnancy. A study done among rural women of Terai region showed that about 29% of women had experienced IPV at some point during their pregnancy [11] whereas, a study done in a hilly district showed that about 91% have experienced some form of gender-based violence during pregnancy [12]. Nepal Demographic and Health Survey, 2011 showed that about 32% of ever-married women reported having experience of any form of physical, sexual and/or emotional violence in their lifetime, and 17% in the past year [13]. 

### 1.3. IPV: Risk Factors and Its Consequences

According to WHO, factors such as; low level of education of women and their partners, having a history of exposure to family violence, harmful use of alcohol of women or their partners, gender inequalities between women and their partners, low level of women’s access to cash earnings, marital controlling behaviors of their partners, and the attitudes of women condoning their partners violence are considered as risk factors of IPV [2]. Numerous studies done globally, including Nepal have used some or many of these variables in examining their effect on domestic violence against women [14,15,16,17,18] showing the mixed relationship between IPV and its correlates. And those arrays of literature were the basis for choosing predictor variables in this study. Violence against women is strongly linked with negative health consequences across their lifespan, such that women who have experienced IPV have more health problems, such as injuries, sexually transmitted diseases, unwanted pregnancies, mental health and other behavioral problems [19]. Among the IPV victims, an estimated 42% of women have experienced injuries resulting from that violence [5]. A systematic review suggests that IPV have an increasingly adverse effect on the psychological well-being of the victims, and the severity and extent of IPV exposure increase the mental health symptoms [20]. Mothers who had experienced IPV had a low utilization of maternal health services [11,21] and was associated with a 51% increase in the risk of pregnancy, and 30% increase in the risk of unwanted pregnancy [22]. Women who are victims of IPV do not enjoy freedom in their relationship. Prior studies that were done in Nepal examining the social norms and women’s risk of IPV contends that Nepalese women have mostly ascribed more conservative gender roles, experienced less agency freedom, and have a restriction on education and employment [23]. Violence against women occurs in all socio-economic classes, but poor women are more likely to experience violence [24]. Majority of women believe that their husbands have the right to beat their wife, and the proportion was more for rural women [24]. 

Despite the growing concern on violence against women, especially by the intimate partner, there is limited understanding regarding the underlying risk factors specified in terms of husband related factors and women’s empowerment characteristics; hence, was the basis for the conception of this study. An attempt is made to analyze the underlying factors based on the broad categories such as; husband related factors, and women empowerment indicators, which would help understand clearly the picture of IPV. The factors which are directly linked to the woman’s husband/partner are categorized as husband related factors, and the factors that are thought to empower the women are categorized as women’s empowerment indicators. Therefore, this study aimed to assess the prevalence and underlying factors of IPV against women aged 15–49 years in Nepal. This study contributes to the existing body of evidence by highlighting the effect of husband’s characteristics and women’s empowerment indicators on IPV.

## 2. Methodology

### 2.1. Study Design, Data, and Sampling

This is an analytical cross-sectional study conducted from the secondary data of Nepal Demographic and Health Survey (NDHS) 2016. NDHS 2016 is a nationally representative survey implemented by New ERA under the support of the Ministry of Health of Nepal, which is the fifth comprehensive cross-sectional survey of its kind conducted as a part of Demographic and Health Surveys (DHS) Program with the technical support by ICF international and financial support of the United States Agency for International Development (USAID) [25]. The dataset was publicly available from the ‘The DHS Program’ website after registering a download account [26]. 

The details of questionnaires and study methodology have been described in the website and survey report [25,26]. In brief, the 2016 NDHS used a multi-stage cluster sampling procedure to collect data wherein each province was stratified into urban and rural areas yielding 14 sampling strata. Then the stratified sample was selected in two and three stages in rural and urban areas respectively. In rural areas, wards were selected as primary sampling units (PSUs) while, in urban areas, one enumerations areas (EAs) was selected from each ward (PSUs). In the first stage, 383 wards were selected using probability proportional method. In an urban area, due to the large size of the urban wards, one EA was selected randomly from each sample wards. For a large cluster, only a segment was selected for the survey so, 2016 NDHS cluster is a ward, an EA, or a segment of a ward or an EA. In the final stage, a fixed number of 30 households per cluster were selected with systematic selection technique. A total of 12,862 women and 4063 men aged 15–49 years from 11,040 households successfully completed the survey with the response rate of 98% and 96% for women and men respectively. 

The 2016 NDHS survey administered six questionnaires: The Household Questionnaire, the Woman’s Questionnaire, the Man’s Questionnaire, the Biomarker Questionnaire, the Fieldworker Questionnaire, and the Verbal Autopsy Questionnaire (for neonatal deaths). The Woman’s Questionnaire was used to collect information on domestic violence from women aged 15–49 years. The 2016 NDHS used a shortened and modified version of the Conflict Tactics Scale to measure domestic violence against women. This study was confined to respondents selected for the domestic violence module from the subsample of households that were selected for the men’s survey. To ensure the privacy, the information collected on domestic violence module from the respondents follows the World Health Organization (WHO) recommended guidelines and ethical standards [27] such that only one eligible woman per household (specifically, the subsample of households that were selected for the men’s survey) was selected randomly for the module. And the domestic violence module was implemented to the eligible women only if privacy could be obtained. A total of 4447 were eligible for the module out of which 4444 successfully completed the face-to-face interview. Of 4444 successfully interviewed women, 882 were never in the union (excluded from the study), 3447 were currently in a union, and 115 were formerly in the union. Our study was restricted to 3562 ever-married women (currently or formerly in a union) and 3447 women (currently in a union) for analyzing an experience of IPV ‘ever’ and ‘in the year preceding the survey’, respectively.

The survey protocol of 2016 NDHS was reviewed and approved by the Nepal Research Health Council (NHRC) and the ICF International Review Board, therefore, an independent ethical review was not needed. 

### 2.2. Measurement of Variables:

#### 2.2.1. Outcome Variables:

The outcome variables for this study is “Experience of IPV” among women aged 15–49 years which was measured by women’s self-reporting of experience of IPV (any of the physical, sexual, or emotional violence). Based on the timeline of experience of IPV, the outcome variable in this study is divided into two types:a)Experience of IPV ‘Ever’ by the current husband or the most recent husband for currently married and formerly married women, respectively.b)Experience of IPV ‘in the year preceding the survey’ by the current husband for currently married women only.

Specifically, the NDHS 2016 measured violence (physical, sexual or emotional violence) committed by their husband (current or former) based on the women’s responses to the following questions asked to them, as shown in Table 1.

This study uses a binary summary measure to capture the experience of IPV, comparing the women who experienced IPV with who have not experienced IPV. Each item has a ‘Yes/No’ response, and value of ‘1′ was given indicating that the act took place (often or sometimes) and the value of ‘0′ was given if the act did not take place. The results of 13 questions were aggregated with the scores ranging from 0–13. The woman was considered ‘not experiencing IPV’ if she scored the value of ‘0′ to in the aggregated score, and a woman with the score of 1 or more was considered as ‘experiencing IPV’. The Cronbach’s alpha of the 13-item scale for an experience of IPV ‘ever’ and ‘in the year preceding the survey’ was 0.899 and 0.895 respectively.

#### 2.2.2. Explanatory Variables:

The definition of the explanatory variables used in this study is shown in Table 2, which were chosen based on the number of existing literature on IPV [2,18,19,28,29,30,31,32]. The household wealth index was evaluated by NDHS 2016 using scores derived from principal component analysis of various household possessions, assets, and amenities [33]. The control variable (socio-demographic variables) included in our models includes women’s age recorded into categories, ethnicity, place of residence, province of residence, wealth status, and women witnessing parental violence. Husband related factors include husband education level, husband alcohol use, marital control behavior displayed by husband, and women being afraid of her husband or not. Women’s empowerment indicators include women’s education, media exposure, cash earnings from their work, ownership of property, decision-making autonomy, attitudes towards the sexual right, attitudes towards wife beating by their husband.

### 2.3. Statistical Analysis

This study has two outcome variables: (a) Experience of IPV ‘Ever’; and (b) experience of IPV ‘in the year preceding the survey’. The data were analyzed with SPSS 25.0. First, data were analyzed using descriptive statistics to describe the characteristics of the study participants and to report the prevalence of experience of IPV ‘Ever’ and ‘in the year preceding the survey’. Secondly, the chi-square(χ^2^) test was used to examine the individual association between experience of IPV ‘ever’ and ‘in the year preceding the survey’, and the independent variables. The significant variables in the chi-square test were then included in the multivariate logistic regression models using hierarchical modeling strategy as done in the previous studies using NDHS data [34]. To conceptualize the analysis, we categorized the explanatory variables into three main groups that could affect the experience of IPV as; (i) socio-demographic characteristics or control variables, (ii) husband characteristics, and (iii) women empowerment indicators. Other studies done in the field of IPV have classified independent variables in different strata [32,35]. In multiple regression analysis, Model 1 was comprised of socio-demographic characteristics (control variables), Model 2 consists of the factors of Model 1 and husband characteristics, and Model 3 consists of the factors of Model 2 and women empowerment indicators. A *p*-value <0.05 was considered as statistically significant. An adjusted odds ratio (AOR), 95% confidence intervals (CIs), and *p*-value were reported. Due to the non-proportional allocation of the sample to their population size in 2016 NDHS, the data needs to be adjusted by sampling weights for analysis. Such sampling weights were provided by the survey to adjust for cluster and strata to ensure that the results are representative at various levels. Based on sample weights, strata and cluster available in the 2016 NDHS dataset, a complex sampling plan file was prepared. All the statistical analysis was then performed using a complex sample analysis procedure which is desired to adjust for sampling weight and multistage sampling procedure in the DHS dataset [36]. There were no multicollinearity issues found in our data when checked for the value of tolerance and variance inflation factor (VIF). 

## 3. Results

### 3.1. Characteristics of the Study Population

Table 3 represents the percentage distribution of the study population by background characteristics. Of the total women, 39.6% were in the age group 35–49 years (mean age/SD; 32.08 yrs./8.64); 30.2% had Brahmin/Chhetri ethnicity; 37.6% were poor; 59.9% and 6.2% were from the urban area and Karnali province respectively. Witnessing parental exposure was low with 85.6% of them reporting to have not witnessed the father ever beating the mother. About 42% of the women and 16.1% of their husband/partner had no formal education, and 18.3% of the women did not have any media exposure at all. Of the total participants, 43.4% were never afraid of their husband; 55.7% and 65.7% reported that their husbands did not drink alcohol and did not show any marital control behavior to them, respectively. About two-thirds (67.4%) did not have any cash earnings from their employment; the majority (81.4%) did not own any property; and, more than one fourth (26.3%) reported to have no participation in household decision-making. The majority (78.7%) of women reported having the autonomy of sexual rights and more than one fourth (29.7%) of women agree that wife beating is justified under some specific circumstances.

### 3.2. Prevalence of Different forms of IPV

Table 4 shows the prevalence of experience of intimate partner violence ‘ever’ and ‘in the year preceding the survey’. Overall, 26.3% and 13.7% of women experienced at least one form of intimate partner violence ‘ever’ and ‘in the year preceding the survey’, respectively. Of the total women who had ever experienced intimate partner violence, 22.8%, 7.0%, and 12.3% had experienced physical, sexual, and emotional violence, respectively. And among the women who had experienced intimate partner violence in the year preceding the survey, 10.1%, 4.0%, and 7.7% had experienced physical, sexual, and emotional violence, respectively. The differences in the overall proportion of IPV and the sum of proportions of different forms of IPV reflects that some women experienced multiple forms of intimate partner violence.

### 3.3. Bivariate Analysis of IPV 

Table 5 shows the experience of intimate partner violence ‘ever’ and ‘in the year preceding the survey’ by explanatory variables. The women of age group 35–49 years had high proportions (28.8%) of ever experience of IPV, whereas, women aged 15–24 years had high proportions (14.5%) of experience of IPV in the year preceding the survey. Women having no formal education had a high prevalence (34.3%) of ever experience of IPV but, the prevalence of experiencing IPV in the year preceding the survey was high among the women who had a primary level of education (16.3%). The prevalence of experience of IPV ‘ever’ and ‘in the year preceding the survey’ both were found more among the women: Of dalit ethnicity (35.5% and 19.3%); in Province 2 (37.1% and 17.3%); of poor wealth status (32.1% and 15.5%); those who witnessing parental violence (45.5% and 23.1%); whose husband does not have formal education (43.6% and 22.4%); whose husband gets drunk very often (73.8% and 43.7%); who is afraid of husband most of the times (73.5% and 46.9%); whose husband shows three or more marital control behavior (74.0% and 54.8%); who does not have any exposure to media at all (32.7% and 17.1%); having no cash earnings (34.0% and 17.9%); having no ownership of property (27.4% and 14.9%); who does not believe in autonomy of sexual rights (37.0% and 22.1%); and who agrees that wife beating is justified for 1 to 2 specific reasons (32.9% and 16.4%), respectively.

Chi-square analysis indicated that women experience of IPV ‘ever’ and ‘in the year preceding the survey’ had significant association with explanatory variables such as; ethnicity, province, witnessing parental violence, husbands education, husband alcohol use, women afraid of their husband, marital control behavior displayed by husband, education of women, women’s cash earnings, women’s ownership of property, women’s attitude towards autonomy of sexual rights, and women’s attitude towards wife beating. Age group (in years), wealth status, and exposure to media were significantly associated with ever experience of IPV, whereas women’s participation in household decision-making was significantly associated with an experience of IPV in the year preceding the survey (Table 5). 

### 3.4. Factors Associated with ‘Ever’ Experiencing IPV

Table 6 shows the complex sample logistic regression analysis of the factors associated with ‘ever’ experience of IPV among women in Nepal. The estimates of Model 1 showed that the odds of ever experiencing IPV were significantly associated with all the sociodemographic factors (control variables): Age group of women (in years), ethnicity, province of residence, wealth status, and women witnessing parental violence. In Model 2, when husband related factors were added to Model 1, all the significant variables in Model 1 except wealth status remained significant and all the husband related factors were significantly associated with ever experience of IPV after controlling for socio-demographic characteristics. In the final model (Model 3), when women empowerment indicators were added to Model 2, all the significant variables in Model 2 remained significant. Of the women empowerment indicators, women cash earnings and women’s attitude towards wife beating were significantly associated with ever experience of IPV after controlling for socio-demographic and husband related factors. In the final model (Model 3), the stronger association of ever experience of IPV was seen among the women whose husband gets drunk very often (AOR: 7.55, CI: 4.68–12.18), who are afraid of their husband most of the times (AOR: 9.36, CI: 5.86–14.93), and whose husband shows 3 or more marital control behaviors (AOR: 9.21, CI: 5.97–14.21). However, women from Brahmin/Chhetri ethnicity (AOR: 0.44, CI: 0.25–0.76) are less likely to report ever experience of IPV.

### 3.5. Factors Associated with ‘Recent’ Experience of IPV

Table 7 shows the complex logistic regression analysis of the factors associated with the experience of IPV ‘in the year preceding the survey’ among women in Nepal. All three socio-demographic factors: Ethnicity, province, and witnessing parental violence were significantly associated with IPV ‘in the year preceding the survey’ (Model 1). When the husband related factors were added in Model 1 (Model 2), witnessing parental violence was the only socio-demographic factors to remain significant. Husband alcohol consumption, women afraid of husband and marital control displayed by husband were the husband related factors that were significantly associated with an experience of IPV ‘in the year preceding the survey’ (Model 2). In the final model (Model 3), when women empowerment indicators are added to Model 2, all the significant variables in Model 2 remained significant, and of the women empowerment indicators, women’s ownership of property, women’s participation in household decision-making and women’s attitude towards autonomy of sexual rights were significantly associated with experiencing IPV ‘in the year preceding the survey’ after controlling for socio-demographic and husband related factors. In the final model (Model 3), the stronger association of experiencing IPV in the year preceding the survey was seen among the women whose husband gets drunk very often (AOR: 3.16, CI: 1.92–5.21), who are most of the time afraid of their husband (AOR: 5.98, CI: 3.74–9.57), and whose husband showed 3 or more marital control behaviors (AOR: 10.64, CI: 7.01–16.16). However, the women who own the property (AOR: 0.59, CI: 0.37–0.94), who participate in 1 to 2 number of household decision making (AOR: 0.61, CI: 0.42–0.87) are less likely to report IPV in the year preceding the survey.

## 4. Discussion

Gender-based violence (or IPV) produces significant public health concerns resulting in physical, sexual and reproductive, and psychological health problems and presents a violation of women’s human rights. In Nepal, the Domestic Violence (Offense and Punishment) Act was passed in 2009, which also includes any acts of violence related to reprimand or emotional abuse [37]. The issues of gender-based violence experienced by women were first examined at the national level in 2011 NDHS [13]. This study examined the gender-based violence (specifically, the Intimate Partner Violence) experienced by women aged 15–49 years by their husband, in Nepal. Many factors, such as husband being uneducated or lower education level, husband alcohol consumption, marital control displayed by husband, women’s cash earnings from employment, women’s attitudes towards wife beating by husband appear to be the risk factors of IPV whereas, higher level of education of husband, women’s ownership of property, and women’s participation in household decision making offered protection against IPV. This study could be an important contribution to the field of IPV in Nepal because of results are based on the national representativeness of the data and provides much attention in exploring the predictors of IPV based on the broad categories of husband related factors and women’s empowerment indicators. 

In our study, we included all forms of partner violence (physical, sexual, and emotional) to measure IPV experienced by women: ‘Ever’; and ‘in the year preceding the survey’. More than one-fourth of the women have reported having ‘ever’ experienced IPV in their lifetime and about 13% have experienced IPV ‘in the past year preceding the survey’. Many studies done in the field of IPV does include only physical, and sexual violence in their study and reported different proportions of IPV [15,21,29]. The actual prevalence may be much higher as most of the women tends to under-report the violence because of social norms, as well as feelings of shame or embarrassment, and stigma attached with discussing marital issues, particularly sex [29]. The victims of domestic violence rarely disclosed their experience of violence, to the health-care personnel or even in a confidential interview [5,38]. Women are less likely to report IPV because of fear of losing social and economic support and ruining the family name [39]. The study done in Kathmandu using the response from the women working in factories reported high proportion of experience of physical IPV (28%), sexual IPV (22%), and emotional IPV (35%) in the past year [40]. The DHS analytical study done using DHS data from various 11 countries showed that Uganda had the highest proportion of women reporting experience of any IPV: ‘Ever’ (58.8%); and ‘in the last 12 months’ (47.0%), whereas, the lowest reporting was found in Tajikistan (ever experience IPV: 23.6%; and in the last 12 months:19.1%) [41]. Of all the forms of IPV, sexual IPV is reported less in Nepal (ever: 7%, in the past 12 months: 4%). The supreme court of Nepal, in 2006 has declared that sex without wife’s consent is punishable by law, and according to the criminal code bill passed by the parliament in 2017, the husband shall be sentenced up to five years in jail for marital rape.

Women aged 35–49 years tend to report more experience of IPV ‘ever’ than women of other age groups. However, an experience of IPV ‘in the past 12 months’ was reported more among the women aged group 15–24 years. Women of younger age group were strongly associated with an increased risk of IPV in the past year [42]. Witnessing parental violence in the family was significantly associated with ever experience of IPV. Odds ratios for IPV was highest where women reported that their mothers had experienced abuse [42]. Therefore, it is important to develop violence prevention programs that target women who have previously witnessed family violence [15]. Women who grew up witnessing parental violence in their family may have more likelihood of accepting violence as a part of everyday life. In our study, we found that higher the household wealth status, lower is the experience of IPV, however, the relation was not significant. In another study, higher socioeconomic status was significantly associated with a decreased risk of IPV [42]. Other studies showed that the household socio-economic status does not have any consistent association with women’s experience of physical violence in the past 12 months [43,44].

Having to live with the husband’s family may create fear for the women because daughters-in-law have little power in the new household and are expected to be subservient in the new family. Therefore, there is always a tendency of “being afraid of someone” in the family which reflects the power imbalance between women and her husband or husband’s families [45]. Majority of the victims of domestic violence reported that they are afraid of someone [38] or husbands in their family [32]. Husband being drunken very often was associated with women’s risk of experiencing IPV ‘ever’ and ‘in the past 12 months’. Similar were the results in the studies where the odds of IPV were higher when the partner had frequent drunkenness [18,42]. However, it is unclear about the causal pathway of which precedes first between alcohol consumption and perpetration of violence: Whether alcohol consumption causes violence, or desire to commit violence leads to alcohol consumption.

Bivariate analysis of education level showed that a reduction in the risk of IPV is associated with higher education of women and her partner. However, in multivariate analysis, only education level of husband/partner was significantly associated with an experience of IPV ‘ever’ and ‘in the past year’. A mixed relationship of IPV was found with an education level of both, women and husband in various studies. A WHO multi-country study on women’s health and domestic violence showed that reduction in IPV risk was associated with secondary education for both the women and her partner, and the highly educated group also had lower odds ratio’s for IPV in 10 out of 14 sites [42]. In another study, neither women’s nor husband’s education were associated with women’s experience of IPV [46]. A study done in India showed that women with lower education level had a higher risk of IPV ‘ever’ and ‘past 12 months’, and the risk of experiencing IPV was higher if their husbands had a lower level of education [47]. However, a study done in China among married rural migrant women showed that a higher level of education of women was associated with a high risk of IPV [44]. But, it is in general understanding that educated women are more likely to marry a more-highly educated man who is less likely to commit violence against women [48]. Higher education not only provides important information for household decision making but also encourages empowerment and autonomy [30]. Similarly, an educated man may value and worth their partner with greater respect. Similar to our findings, the study done in Nigeria showed that access to media did not have any significant relationship with women being the victims of IPV [29].

Our study showed that the women who received cash earnings from their work were more likely to report IPV at some point in their lives, but the association was not significant for the ‘past year’ experience of IPV. A multi-country study showed that working women with an unemployed partner were at higher risk of IPV whereas, unemployed women with an employed partner tend to be at lower risk of IPV, and if neither of them was employed, the risk was higher [42]. However, Ackerson LK et.al. argues that higher education level of women provides more opportunities for financial independence and thereby providing the husband with an enticement that refrains him from abusing her [47]. In contrast to our findings, women with low financial autonomy in the past 12 months were significantly associated with higher levels of IPV [44]. Lack of economic equality in any relationship may serve as a predictor of IPV rather than household socio-economic status does.

Higher the marital control behavior displayed by husband, higher is the risk of experiencing IPV lifetime and recent (past 12 months). The proportion of women reporting one or more acts of controlling behavior by their partner is normative to different degrees in various settings, ranging from 21% in Japan city to 90% in Tanzania city [7]. And, women who suffered IPV reported more acts of controlling behavior by their intimate partner [7,32,49]. A qualitative study done in India showed that men abuses their wives to vent their frustrations which they could not exhaust in public, and in addition, they use violence to proclaim their authority over women [50]. Husbands with a lower level of education are more likely to believe that they are justified in displaying marital control behavior to their wives and in using physical force achieve this dominion [51]. The women who agree that the husband beating their wife is justified are at increased risk of experiencing IPV. The women who had attitudes supportive to wife beating by husband had increased risk of IPV [42]. These justifying attitudes reflect the social norms of gender inequality that privilege men to have power over women. It is necessary to change the gender norms and social stigma that women should remain inert when male counterpart shows controlling behavior. 

Women who believe in the autonomy of their sexual rights were more likely to report an experience of IPV in the past 12 months. The permissive social norms and beliefs about women’s autonomy on sexual rights have significant independent and incremental effect on risk of experiencing IPV in the past year, so one needs to understand the social norms before developing any programs that address domestic violence [29]. However, in contrast, a study showed that women’s ability to decide when to have sexual relations reduce the risk of violence [43]. Both men and women should be made aware of the women’s autonomy on sexual rights. Women who participate in household decision making were less risk of experiencing IPV in the past 12 months. Lower the women’s ability to participate in household decision making, higher is the risk of experiencing IPV [15,52,53]. A study done in the Philippines showed a U-shaped relationship of higher levels of IPV with both, husbands dominating and wives dominating in decision making, however, lower levels of IPV was associated with joint decision making [54]. Joint decision making by the couple is an important pre-condition for better spousal understandings [32]. In the Nepalese context, empowering women through participation in household-decision making is a first step intervention on a policy level [15]. This shows that women are less likely to experience IPV when the couple makes a joint decision. Empowering women to participate in household-related decision-making may prevent them from being victims of IPV. 

### 4.1. Strength and Limitations of the Study

This study is based on the data from the large survey at the national level with a high response rate. And the sampling weights are used to adjust the multi-stage sampling procedure of the survey so that this data can be made nationally representative. We classified variables into three broad categories: Sociodemographic, husband related, and women’s empowerment related factors and developed the models accordingly so that this study provides valuable information regarding factors affecting IPV which could help policymakers and activist to program interventions to address domestic violence. However, there are some limitations to this study. Since this survey is cross-sectional in nature, the temporal relationship between covariates and outcomes cannot be established. Also, since domestic violence is a sensitive issue and social stigma is attached to it, respondents may be hesitant to report their actual experience of IPV, thus there can be some social desirability bias or reporting bias. We used an aggregated score of IPV (physical, sexual, and emotional) as an outcome measure in our study instead of studying different forms of violence independently. We also believe that domestic violence being a normative measure, can be addressed appropriately with qualitative data than quantitative data. Therefore, there is a need for further studies focusing on qualitative research design to display a good understanding of domestic violence. 

### 4.2. Implications of the Study

The multi-dimensional nature of the factors that affect IPV and the identified risk factors highlights the need for a multi-sectoral approach and a comprehensive intervention for the prevention of IPV. Not all the variables significant in bivariate analysis demonstrated a consistent relationship with IPV in multivariate analysis. Also, the variable significant in ‘ever’ experience of IPV was not found to be significant when the outcome was ‘past year’ experience of IPV. Therefore, policymakers and activist should be vigilant about any “one model fits all” approach when formulating policies to prevent IPV. As it is seen from the findings that the access to resources for empowering women do not necessarily decrease the risk of violence against them. For example, a higher education level of women did not significantly associate with lower risk of IPV. Also, the higher level of education of the husband does not necessarily guarantee them from refraining in violence against women. Therefore, it is important that prevention efforts and strategies should be engaged with both men and women. In addition, media and advocacy campaigns should be organized to raise awareness about existing legislation on domestic violence.

## 5. Conclusions

This study revealed that more than one-fourth of women, and about 14 percent of women reported lifetime, and recent (past 12 months) experience of any type of IPV, respectively. Most of the women reported physical IPV ‘ever’ and ‘in the past 12 months’. Most of the women had experienced multiple forms of violence. The risk factors were different based on either ‘lifetime’ or ‘past 12 months’ experience of violence. Women witnessing parental violence during their childhood, the husband being drunken frequently, women being afraid of their husband most of the times, women whose husbands shows marital control behaviors were found to be significantly associated with an experience of IPV, both ‘lifetime’ and ‘in the past 12 months’. However, women of older age group, husband with lower education, women who has cash earnings, and who believes that wife beating by their husband is justified are more likely to report lifetime experience of IPV, whereas, women who own the property, who participates in household decision making are less likely to report recent experience of IPV. Husband related factors were found to have a stronger significant association with experience of IPV than with women’s empowerment indicators. A complex blend of social norms related to gender, family primacy and attitudes towards acceptability of violence undergird the perpetration of IPV. Therefore, reducing IPV requires a commitment to changing the norms that promote the husband’s behavior of controlling his wives and beating her. And the socially acceptable norms and values that promote mutual respect between couples should be emphasized and should also be communicated to the new generations. The interventions to address violence against women requires the involvement and coordination of various actors working together at the community, state, and national levels. 

## Figures and Tables

**Table 1 ijerph-16-00709-t001:** Measurement of outcome variables.

Forms of Violence	Measurements (Questions Asked to Women if Their Husband Did the Following Events)
Physical violence (seven questions)	a. Pushed, shook or thrown something at her; b. Slapped her; c. Twisted arm or pulled her hair; d. Punched her with a fist or something that could hurt her; e. Kicked, dragged or beat her; f. Tried to choke or burn her on purpose; and, g. Threatened or attacked her with any weapon, such as a knife, gun or any other weapon.
Sexual violence (three questions)	h. Physically forced to have unwanted sexual relationships with him; i. Physically forced to perform any other unwanted sexual acts; and, j. Forced with threats and any other way to perform unwanted sexual acts
Emotional violence (three questions)	k. Humiliated her in front of others; l. Threatened to hurt or harm her or someone close to her; and, m. Insulted or made her feel bad about herself.

**Table 2 ijerph-16-00709-t002:** Measurement of explanatory variables.

Variables	Measurement
Age group (in years)	Self-reported age of women at the time of the survey, grouped into 15–24 years; 25–34 years; and 35–49 years
Ethnicity	Self-reported ethnic affiliation of respondents grouped into Brahmin/Chhetri (Hill Brahmin/Chhetri, Terai Brahmin/Chhetri); Janajati (Newar, Hill/Terai Janajati); Dalit (Hill/Terai Dalit); and Other castes (all other ethnicities)
Place of residence	Types of a place of residence: Urban; and Rural
Province	The provincial residence of respondent at the time of the survey: Province 1; Province 2; Province 3; Gandaki Province; Province 5; Karnali Province; Sudurpaschim Province
Household wealth status	A composite index of household possessions, assets, and amenities, derived using principal component analysis, grouped as, Poor (Poorest and Poorer); Middle; and Rich (Richer and Richest)
Witnessing parental violence	Self-reported history of witnessing violence in the family measured as, Father ever beat her mother: Yes; No
Husband/Partner education	The highest level of education attained by husband/partner: No Education; Primary; Secondary; Higher
Husband/Partner alcohol use	Respondent reporting of partner’s frequency of alcohol consumption, measured as; Does not drink; Drinks but never get drunk; Gets drunk sometimes; Gets drunk very often
Women afraid of husband	Self-reported behavior of women being afraid of their husband/partner as; Never afraid; Sometimes afraid; Most of the time afraid
Marital control behavior displayed by husband	A composite variable reflecting respondent self-reporting of five controlling behavior displayed by the husband/partner (is jealous if she talks to other men; accuses respondent of being unfaithfulness; does not permit respondent to meet female friends; tries to limit respondent’s contact with family; insists on knowing where respondent was), grouped into: No behavior displayed; 1–2 behavior displayed; 3 or more behavior displayed
Education of women	The highest level of education attained by respondents: No Education; Primary; Secondary; Higher
Exposure to media	A composite variable derived from the frequency of access to newspaper/magazine, radio and television, grouped as, No exposure; Exposure to 1–2 media; Exposure to all 3 media
Women’s cash earnings	Self-report of types of earning from respondent’s work, grouped into, No cash earnings (Not paid and In-kind only); Cash earnings (Cash only and/or Cash and in-kind)
Ownership of property	A composite variable derived from the respondent’s ownership of house, land or both alone or jointly with husband, grouped as: Does not own (Does not own at all); Owns a property (Owns house, land or both alone or jointly with husband)
Women’s participation in household decision making	A composite variable measured from women’s participation (alone or with husband) in making three household decisions (access to health care; large household purchases; and freedom to visit families and relatives), grouped into, No participation; Participation in 1–2 decision making; Participation in all 3-decision making
Attitudes towards the autonomy of sexual rights	A composite score of women’s abilities to negotiate sexual relations with husband measured from responses of two questions: Women can refuse sex if they don’t want; and can ask their husband to use a condom. The score ranges between 0 and 2, measured as attitudes towards the autonomy of women’s sexual rights: Accepts sexual right (score of 2); Does not believe in sexual rights (score of 0 and 1)
Attitudes towards wife beating (no. of reasons for which wife beating is justified)	A composite variable reflecting women’s attitudes towards wife beating by their husband for each of the following five reasons (goes out without telling her husband; neglects the children; argues with husband; refuses to have sex with husband; and burns the food), grouped as: Not justified; Justified for 1–2 reasons; Justified for 3–5 reasons

**Table 3 ijerph-16-00709-t003:** Descriptive characteristics of the study population from the 2016 Nepal Demographic and Health Survey (NDHS) (*n* = 3562).

Variables	Categories	Number #	Percentage #
Age group (in years)	15–24	832	23.4
	25–34	1318	37.0
	35–49	1412	39.6
	Mean/S.D.	32.08	8.64
Ethnicity	Brahmin/Chhetri	1076	30.2
	Janajati (Indigenous)	1274	35.8
	Dalit	479	13.4
	Other castes	734	20.3
Place of residence	Rural	1429	40.1
	Urban	2133	59.9
Province	Province 1	597	16.7
	Province 2	782	21.9
	Province 3	679	19.1
	Gandaki Province	353	9.9
	Province 5	618	17.3
	Karnali Province	222	6.2
	Sudurpaschim Province	312	8.8
Household wealth status	Poor	1341	37.6
	Middle	756	21.2
	Rich	1465	41.1
Witnessing parental violence	No	3048	85.6
	Yes	514	14.4
Husband/Partner education	No education	555	16.1
	Primary	766	22.2
	Secondary	1508	43.8
	Higher	613	17.8
Husband/Partner alcohol use	Does not drink	1984	55.7
	Drinks but never get drunk	437	12.3
	Gets drunk sometimes	881	24.7
	Gets drunk very often	260	7.3
Women afraid of husband	Never	1547	43.4
	Sometimes	1745	49.0
	Most of the times	270	7.6
Marital control behavior displayed by husband	No behavior displayed	2341	65.7
	1–2 behavior displayed	907	25.5
	3 or more behavior	314	8.8
Education of women	No education	1491	41.9
	Primary	667	18.7
	Secondary	1000	28.1
	Higher	405	11.4
Exposure to media	No exposure	654	18.3
	Exposure to 1–2 media	2300	64.6
	Exposure to all 3 media	609	17.1
Women’s cash earnings	No cash earnings	2401	67.4
	Cash earnings	1162	32.6
Ownership of property	Does not own	2899	81.4
	Owns a property	664	18.6
Women’s participation in household decision making	No participation	906	26.3
	1–2 decisions	1193	34.6
	All 3 decisions	1348	39.1
Attitude towards the autonomy of sexual rights	Does not believe	733	21.3
	Accepts sexual rights	2714	78.7
Attitudes towards wife beating (no. of reasons for which wife beating is justified)	Not justified	2505	70.3
	Justified for 1–2 reasons	806	22.6
	Justified for 3–5 reasons	252	7.1

# Number and percentage are adjusted for the multi-stage sampling, cluster weight, and sampling weight. S.D: Standard Deviation.

**Table 4 ijerph-16-00709-t004:** Prevalence of experience of intimate partner violence.

Forms of Violence	Experience of IPV Ever (*n* = 3562)		Experience of IPV in the Year Preceding the Survey (*n* = 3447)	
	Number #	Percentage #	Number #	Percentage #
Physical IPV	812	22.8	347	10.1
Sexual IPV	251	7.0	137	4.0
Emotional IPV	438	12.3	265	7.7
Any IPV (Either Emotional or Sexual or Physical)	938	26.3	471	13.7

# Number and Percentage is adjusted for the multi-stage sampling, cluster weight, and sampling weight.

**Table 5 ijerph-16-00709-t005:** Experience of intimate partner violence ‘ever’ and ‘in the year preceding the survey’ by background characteristics.

Variables	Categories	Ever Experience IPV (*n* = 3562)				Experience of IPV in the Year Preceding the Survey (*n* = 3447)			
		N #	% #	χ^2^-Value	*p*-Value	N #	% #	χ^2^-Value	*p*-Value
**Socio-Demographic Characteristics**									
Age group (in years)	15–24	180	21.6	15.34	0.015 *	119	14.5	1.33	0.720
	25–34	351	26.6			181	14.0		
	35–49	407	28.8			172	12.9		
Ethnicity	Brahmin/Chhetri	170	15.8	161.54	<0.001 *	91	8.8	45.82	<0.001 *
	Janajati (Indigenous)	308	24.2			166	13.4		
	Dalit	170	35.5			89	19.3		
	Other castes	290	39.5			125	17.6		
Place of residence	Rural	396	27.7	2.48	0.311	206	15.0	3.59	0.154
	Urban	542	25.4			265	12.8		
Province	Province 1	129	21.6	93.34	<0.001 *	58	10.0	31.56	0.001 *
	Province 2	290	37.1			133	17.3		
	Province 3	176	25.9			102	15.6		
	Gandaki	55	15.5			27	7.9		
	Province 5	178	28.8			90	15.2		
	Karnali	42	19.1			26	12.2		
	Sudurpaschim	67	21.6			35	11.7		
Household wealth status	Poor	357	26.6	22.89	0.003 *	189	14.7	7.62	0.131
	Middle	243	32.1			113	15.5		
	Rich	338	23.1			170	11.9		
Witnessing parental violence	No	704	23.1	121.67	<0.001 *	359	12.1	46.06	<0.001 *
	Yes	234	45.5			112	23.1		
**Husband Characteristics**									
Husband/partner education	No education	242	43.6	192.38	<0.001 *	124	22.4	71.07	<0.001 *
	Primary	244	31.9			126	16.5		
	Secondary	316	20.9			174	11.5		
	Higher	79	13.0			46	7.6		
Husband/partner alcohol use	Does not drink	355	17.9	425.70	<0.001 *	170	8.8	258.13	<0.001 *
	Drinks/never get drunk	102	23.2			40	9.4		
	Gets drunk sometimes	290	32.9			159	18.5		
	Gets drunk very often	192	73.8			101	43.7		
Women afraid of husband	Never	208	13.4	492.39	<0.001 *	88	5.9	337.66	<0.001 *
	Sometimes	531	30.5			267	15.7		
	Most of the times	199	73.5			116	46.9		
Marital control behavior displayed by husband	No behavior	350	14.9	647.49	<0.001 *	142	6.2	590.28	<0.001 *
	1–2 behavior	356	39.2			170	19.1		
	3 or more behavior	232	74.0			161	54.8		
**Women Empowerment Characteristics**									
Education of women	No education	511	34.3	129.68	<0.001 *	226	15.8	24.81	0.026 *
	Primary	190	28.6			104	16.3		
	Secondary	184	18.5			104	10.6		
	Higher	52	12.8			38	9.4		
Exposure to media	No exposure	214	32.7	35.32	<0.001 *	106	17.1	14.19	0.072
	Any 2 media	612	26.6			307	13.7		
	All 3 media	113	18.5			58	9.9		
Women’s cash earnings	No cash earnings	543	22.6	55.78	<0.001 *	274	11.7	26.00	<0.001 *
	Cash earnings	395	34.0			197	17.9		
Ownership of property	Does not own	793	27.4	9.30	0.015 *	421	14.9	22.59	<0.001 *
	Owns a property	145	21.8			50	8.0		
Women’s participation in household decision making	No participation	238	26.3	2.53	0.436	157	17.3	15.10	0.011 *
	1–2 decisions	287	24.0			142	11.9		
	All 3 decisions	357	26.5			172	12.8		
Attitude towards the autonomy of sexual rights	Does not believe	271	37.0	68.29	<0.001 *	162	22.1	59.57	<0.001 *
	Accepts sexual rights	611	22.5			310	11.4		
Attitudes towards wife beating (no. of reasons for which wife beating is justified)	Not justified	593	23.7	32.94	<0.001 *	306	12.6	9.27	0.043 *
	1–2 reasons	265	32.9			126	16.4		
	3–5 reasons	79	31.5			40	16.1		

N: Number, %: Percentage, χ^2^: Chi-square, * reflects statistically significant association in Chi-square test, IPV: Intimate Partner Violence, # The number and percentage are adjusted for multi-stage sampling, cluster weight, and sample weight.

**Table 6 ijerph-16-00709-t006:** Analysis of factors associated with Ever experience of IPV among women in Nepal, 2016 NDHS.

Variables	Categories	Ever Experience IPV; Adjusted OR (95% CI)		
		Model 1	Model 2	Model 3
**Socio-Demographic Characteristics**				
Age group (in years)		***p* = 0.001**	***p* = 0.003**	***p* = 0.028**
	15–24	1	1	1
	25–34	1.53 (1.15–2.05)	1.57 (1.14–2.16)	1.46 (1.05–2.04)
	35–49	1.69 (1.29–2.21)	1.79 (1.28–2.52)	1.67 (1.13–2.46)
Ethnicity		***p* < 0.001**	***p* = 0.019**	***p* = 0.027**
	Other castes	1	1	1
	Brahmin/Chhetri	0.32 (0.21–0.48)	0.44 (0.26–0.75)	0.44 (0.25–0.76)
	Janajati (Indigenous)	0.50 (0.33–0.74)	0.47 (0.28–0.78)	0.46 (0.26–0.78)
	Dalit	0.83 (0.56–1.21)	0.64 (0.37–1.08)	0.61 (0.36–1.03)
Province		***p* = 0.006**	***p* = 0.004**	***p* = 0.015**
	Gandaki	1	1	1
	Karnali	1.36 (0.89–2.08)	1.29 (0.83–2.01)	1.19 (0.76–1.87)
	Sudurpaschim	1.65 (1.09 - 2.51)	2.25 (1.40–3.62)	2.02 (1.24–3.28)
	Province 1	1.44 (0.96–2.15)	1.40 (0.89–2.19)	1.34 (0.85–2.13)
	Province 2	2.00 (1.26–3.19)	1.52 (0.87–2.67)	1.44 (0.81–2.56)
	Province 3	2.04 (1.35–3.08)	2.01 (1.26–3.22)	1.77 (1.10–2.84)
	Province 5	2.02 (1.31–3.10)	2.08 (1.27–3.42)	2.03 (1.24–3.31)
Household wealth status		***p* = 0.001**	*p* = 0.571	*p* = 0.804
	Rich	1	1	1
	Middle	1.53 (1.16–2.02)	1.17 (0.84–1.61)	1.11 (0.79–1.56)
	Poor	1.48 (1.17–1.87)	1.00 (0.73–1.37)	1.01 (0.72–1.43)
Witnessing parental violence		***p* < 0.001**	***p* < 0.001**	***p* < 0.001**
	No	1	1	1
	Yes	2.87 (2.22–3.70)	2.64 (1.92–3.64)	2.64 (1.90–3.66)
**Husband Characteristics**				
Husband/partner education			***p* < 0.001**	***p* = 0.004**
	Higher		1	1
	Secondary		1.47 (0.97–2.23)	1.46 (0.97–2.21)
	Primary		2.02 (1.28–3.17)	1.89 (1.19–3.01)
	No education		2.90 (1.76–4.78)	2.46 (1.47–4.14)
Husband/partner alcohol use			***p* < 0.001**	***p* < 0.001**
	Does not drink		1	1
	Drinks/never get drunk		1.70 (1.17–2.47)	1.71 (1.17–2.48)
	Gets drunk sometimes		2.04 (1.51–2.76)	2.04 (1.50–2.78)
	Gets drunk very often		7.66 (4.75–12.35)	7.55 (4.68–12.18)
Women afraid of husband			***p* < 0.001**	***p* < 0.001**
	Never		1	1
	Sometimes		2.52 (1.94–3.27)	2.49 (1.91–3.24)
	Most of the times		9.65 (6.05–15.47)	9.36 (5.86–14.93)
Marital control behavior displayed by husband			***p* < 0.001**	***p* < 0.001**
	No behavior		1	1
	1–2 behavior		2.98 (2.27–3.92)	2.87 (2.18–3.78)
	3 or more behavior		9.65 (6.23–14.93)	9.21 (5.97–14.21)
**Women Empowerment Characteristics**				
Education of women				*p* = 0.824
	Higher			1
	Secondary			0.92 (0.56–1.50)
	Primary			1.00 (0.58–1.74)
	No education			1.12 (0.65–1.94)
Exposure to media				*p* = 0.840
	No exposure			1
	Any 2 media			1.09 (0.81–1.46)
	All 3 media			1.10 (0.70–1.73)
Women’s cash earnings				***p* = 0.004**
	No cash earnings			1
	Cash earnings			1.49 (1.13–1.95)
Ownership of property				*p* = 0.271
	Does not own			1
	Owns a property			0.83 (0.60–1.15)
Attitude towards the autonomy of sexual rights				*p* = 0.288
	Does not believe			1
	Accepts sexual rights			1.15 (0.88–1.51)
Attitudes towards wife beating (no. of reasons for which wife beating is justified)				***p* = 0.018**
	Not justified			1
	1–2 reasons			1.52 (1.14–2.04)
	3–5 reasons			1.06 (0.71–1.56)
Nagelkerke’s R-square		0.125	0.411	0.421

Model 1: Age group, ethnicity, province, household wealth status, and witnessing parental violence. Model 2: Husband/Partner education, husband/partner alcohol use, women afraid of husband, marital control behavior displayed by the husband. Model 3: Education of women, exposure to media, women’s cash earnings, ownership of property, attitude towards the autonomy of sexual rights, and attitude towards wife beating. 1–reference category, *p* = *p*-value of the variables obtained from the test of model effects, IPV: Intimate Partner Violence, OR: Odds Ratio, CI: Confidence Interval. All values are weighted for the multi-stage sampling, cluster weight, and sampling weight.

**Table 7 ijerph-16-00709-t007:** Analysis of factors associated with experiencing IPV in the year preceding the survey among women in Nepal, 2016 NDHS.

Variables	Categories	Experience of IPV in the Year Preceding the Survey; Adjusted OR (95% CI)		
		Model 1	Model 2	Model 3
**Socio-Demographic Characteristics**				
Ethnicity		***p* < 0.001**	*p* = 0.560	*p* = 0.842
	Other castes	1	1	1
	Brahmin/Chhetri	0.49 (0.30–0.79)	0.74 (0.43–1.28)	0.82 (0.47–1.42)
	Janajati (Indigenous)	0.76 (0.50–1.16)	0.78 (0.46–1.30)	0.81 (0.48–1.36)
	Dalit	1.14 (0.71–1.82)	0.97 (0.54–1.73)	0.95 (0.53–1.68)
Province		***p* = 0.031**	*p* = 0.057	*p* = 0.092
	Gandaki	1	1	1
	Karnali	1.83 (1.10–3.05)	1.62 (0.98–2.69)	1.50 (0.90–2.51)
	Sudurpaschim	1.77 (1.05–2.99)	1.84 (1.01–3.33)	1.64 (0.92–2.93)
	Province 1	1.30 (0.79–2.16)	1.29 (0.75–2.23)	1.30 (0.75–2.27)
	Province 2	1.92 (1.11–3.33)	1.30 (0.69–2.44)	1.19 (0.65–2.20)
	Province 3	2.13 (1.31–3.45)	2.17 (1.30–3.62)	2.05 (1.23–3.45)
	Province 5	1.99 (1.23–3.23)	1.87 (1.13–3.07)	1.74 (1.06–2.84)
Witnessing parental violence		***p* < 0.001**	***p* = 0.015**	***p* = 0.013**
	No	1	1	1
	Yes	2.04 (1.47–2.83)	1.70 (1.10–2.61)	1.68 (1.11–2.55)
**Husband Characteristics**				
Husband/partner education			*p* = 0.117	*p* =0.054
	Higher		1	1
	Secondary		1.25 (0.67–2.34)	1.43 (0.87–2.35)
	Primary		1.39 (0.75–2.59)	1.66 (1.00–2.76)
	No education		1.93 (0.97–3.81)	2.27 (1.22–4.23)
Husband/partner alcohol use			***p* < 0.001**	***p* < 0.001**
	Does not drink		1	1
	Drinks/never get drunk		1.21 (0.75–1.95)	1.26 (0.79–2.01)
	Gets drunk sometimes		1.89 (1.31–2.70)	2.00 (1.41–2.84)
	Gets drunk very often		3.06 (1.86–5.05)	3.16 (1.92–5.21)
Women afraid of husband			***p* < 0.001**	***p* < 0.001**
	Never		1	1
	Sometimes		2.38 (1.63–3.46)	2.30 (1.58–3.33)
	Most of the times		6.43 (3.99–10.38)	5.98 (3.74–9.57)
Marital control behavior displayed by husband			***p* < 0.001**	***p* < 0.001**
	No behavior		1	1
	1–2 behavior		2.87 (2.04–4.04)	2.75 (1.97–3.83)
	3 or more behavior		10.71 (7.18–15.98)	10.64 (7.01–16.16)
**Women Empowerment Characteristics**				
Education of women				*p* = 0.179
	Higher			1
	Secondary			0.67 (0.35–1.28)
	Primary			0.71 (0.34–1.49)
	No education			0.56 (0.31–1.01)
Women’s cash earnings				*p* = 0.057
	No cash earnings			1
	Cash earnings			1.38 (0.99–1.93)
Ownership of property				***p* = 0.028**
	Does not own			1
	Owns a property			0.59 (0.37–0.94)
Women’s participation in household decision making				***p* = 0.023**
	No participation			1
	1–2 decisions			0.61 (0.42–0.87)
	All 3 decisions			0.68 (0.46–1.00)
Attitude towards the autonomy of sexual rights				***p* = 0.002**
	Does not believe			1
	Accepts sexual rights			1.58 (1.19–2.11)
Attitudes towards wife beating (no. of reasons for which wife beating is justified)				*p* = 0.618
	Not justified			1
	1–2 reasons			1.17 (0.84–1.63)
	3–5 reasons			1.09 (0.65–1.83)
Nagelkerke’s R-square		0.049	0.320	0.340

Model 1: Ethnicity, province, and witnessing parental violence. Model 2: Husband/Partner education, husband/partner alcohol use, women afraid of husband, marital control behavior displayed by the husband. Model 3: Education of women, women’s cash earnings, ownership of property, women’s participation in household decision making, attitude towards the autonomy of sexual rights, and attitude towards wife beating. 1—reference category, *p* = *p*-value of the variables obtained from the test of model effects, IPV: Intimate Partner Violence, OR: Odds Ratio, CI: Confidence Interval. All values are weighted for the multi-stage sampling, cluster weight, and sampling weight.

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
