# Peer review of "Intimate Partner Violence in Relation to Husband Characteristics and Women Empowerment: Evidence from Nepal"

_ijerph, 2019, doi:10.3390/ijerph16050709_

Round 1

Reviewer 1 Report

This is a well-written paper on IPV and partner characteristics in Nepal. I have a few suggestions for how to strengthen it for publication.

The abstract says IPV incidence is more related to husband characteristics than women's empowerment indicators, but the results show that both are important.

P2 lines 45-48 could be shortened to one key statistic about the East Asian region.

Are any of the covariates dyadic in nature - for example comparing the husband's education level to that of the woman? If not, explain why the dyadic nature of the data were not harnessed.

Would report tables as a single row whenever possible (i.e. "Owns property vs not")

In line 374, reiterate what this study found with regards to employment status.

How do these findings extend beyond what has already been shown in many other studies globally? It's clear how these data match what is known in the IPV field, but not how they add to the literature.

Limitations should include reporting bias, since earlier in Introduction the authors purport that women rarely disclose incidents of violence even in confidential interviews.

Author Response

Dear Reviewer,

Please find the responses to the reviewer's comments and suggestions as an attached file.

Thanking you for your co-operation.

Reviewer 2 Report

Intimate Partner Violence in Relation to Husband Characteristics and Women Empowerment: Evidence from NDHS 2016

A brief summary

This article presents the findings from the national, cross-sectional Nepal Demographic and Health Survey 2016. A range of secondary data analysis strategies were used to explore the factors that might relate to intimate partner violence against women.  The findings show husband characteristics those related to women empowerment have different associations depending upon if IPV experience was ‘ever’ or ‘in the last 12 months’.  The article concludes that social norms around gender relations need to change in order to address the relatively high incidence of IPV.

Broad comments

A very important topic and one that needs greater attention.  It is also great to see this kind of data from a developing country such as Nepal.  Overall, the article could be improved with better use of summaries, paragraphs and general editing.  The most important aspect of the article, that is the implications and impacts of the wider.  Given the focus of the article more description about what is meant by ‘husband factors’ and ‘women empowerment’ is needed at the outset within the introduction in order to connect better to the discussion.  The results are well laid out.  The discussion section has a lot of great information in it but it could be written a lot more clearly.  It would benefit from a summary at the beginning highlighting the main issues that will be discussed.  Perhaps using headings to sign post the main issues would also help with making the main issues more prominent.

Specific comments

Para beginning line 60 Is the information discussed in this para what has been found in other literature?  It needs to be made clearer.

Line 62  attitudes of who?

Line 63 Numerous literature – globally or still from Nepal? 

Para beginning line 74   make it clearer if you are meaning other literature shows these things mentioned here.

Para beginning 79 In this paragraph the contribution of this article needs to made more clearly.  It would also be useful here to describe what you mean by ‘husband related factors’ and women empowerment indicators” as they are central ideas to your analysis.

Bullet points beginning line 133  The information bulleted would possibly work better as a table.

Line 311-312  Explain more clearer what you mean in this last sentence

Author Response

(The authors gave the same response as above.)

Round 2

Reviewer 1 Report

All suggested edits have been addressed to satisfaction.

Author Response

Dear Reviewer, Please find the responses to the comments and suggestions as an attached file.

Thank you.
